

# Effect of oral tryptamines on the gut microbiome of rats—a preliminary study

Mengyang Xu[1], Andor J. Kiss[2], J. Andrew Jones[3], Matthew S. McMurray[4] and Haifei Shi[1]

[1] Biology, Miami University, Oxford, OH, United States
[2] Center for Bioinformatics and Functional Genomics, Miami University, Oxford, OH, United States
[3] Chemical, Paper, and Biomedical Engineering, Miami University, Oxford, OH, United States
[4] Psychology, Miami University, Oxford, OH, United States

## ABSTRACT

**Background:** Psilocybin and related tryptamines have come into the spotlight in recent years as potential therapeutics for depression. Research on the mechanisms of these effects has historically focused on the direct effects of these drugs on neural processes. However, in addition to such neural effects, alterations in peripheral physiology may also contribute to their therapeutic effects. In particular, substantial support exists for a gut microbiome-mediated pathway for the antidepressant efficacy of other drug classes, but no prior studies have determined the effects of tryptamines on microbiota.

**Methods:** To address this gap, in this preliminary study, male Long Evans rats were treated with varying dosages of oral psilocybin (0.2 or 2 mg/kg), norbaeocystin (0.25 or 2.52 mg/kg), or vehicle and their fecal samples were collected 1 week and 3 weeks after exposure for microbiome analysis using integrated 16S ribosomal DNA sequencing to determine gut microbiome composition.

**Results:** We found that although treatment with neither psilocybin nor norbaeocystin significantly affected overall microbiome diversity, it did cause significant dose- and time-dependent changes in bacterial abundance at the phylum level, including increases in *Verrucomicrobia* and *Actinobacteria*, and decreases in *Proteobacteria*.

**Conclusion and Implications:** These preliminary findings support the idea that psilocybin and other tryptamines may act on the gut microbiome in a dose- and time-dependent manner, potentially identifying a novel peripheral mechanism for their antidepressant activity. The results from this preliminary study also suggest that norbaeocystin may warrant further investigation as a potential antidepressant, given the similarity of its effects to psilocybin.

Corresponding authors
Matthew S. McMurray,
mcmurrms@miamioh.edu
Haifei Shi, shih@miamioh.edu

## INTRODUCTION

In addition to its hallucinogenic properties, psilocybin has gained recent interest as a potential fast-acting treatment for depression (*Nichols, 2004*). A growing number of clinical studies have suggested that when paired with talk therapy, a single dose of

psilocybin can have strong and persistent effects that may be equal to or greater than traditional antidepressants, including widely used selective serotonin reuptake inhibitors, such as fluoxetine (Prozac), sertraline (Zoloft), and escitalopram (Lexapro), for treating major depressive disorder (*Becker et al., 2022*; *Carhart-Harris et al., 2021*; *Carhart-Harris et al., 2018*; *Davis et al., 2021*; *Goodwin et al., 2022*; *Goodwin et al., 2023a*, *2023b*; *Griffiths et al., 2016*; *Gukasyan et al., 2022*). The mechanisms of these effects have not yet been fully understood. Previous findings have implied that psilocybin's psychedelic and hallucinogenic properties depend on a mixture of actions on excitatory and inhibitory neuronal circuits, with substantial evidence pointing to its activation of 5-HT$_{2A}$ receptors as a likely mechanism (*De Gregorio et al., 2021*; *González-Maeso et al., 2007*; *Presti & Nichols, 2004*). However, the mechanisms of its therapeutic effects are still unclear and may be independent from its hallucinogenic effects. In animal models, co-treatment with ketanserin, a 5-HT$_{2A}$ antagonist, has been shown to block both the psychedelic effects of psilocybin and its therapeutic efficacy (*Hesselgrave et al., 2021*; *Slocum, DiBerto & Roth, 2022*). Additionally, animal models have shown that selective activation of 5-HT$_{2A}$ receptors by other compounds can recapitulate many of the psychedelic effects of psilocybin (*Hanks & González-Maeso, 2013*). This mechanism differs from conventional antidepressants such as selective serotonin reuptake inhibitors (SSRIs) that block serotonin transporters and elevate serotonin levels in neuronal synapses. With these traditional antidepressants, despite acutely increasing in serotonin levels, phenotypic depressive symptoms are usually not relieved until 4–6 weeks later (*Stahl, 2021*).

Much of the research investigating the mechanisms of psilocybin and other antidepressants has focused on their interaction with central nervous system processes (*Chen et al., 2023*; *Meccia, Lopez & Bagot, 2023*), including those reviewed above. However, a growing body of research has pointed to alternative mechanisms for antidepressant action, through peripheral processes. Specifically, significant research indicates that modulation of gut processes such as gut motility, permeability, and gut microbiome composition may be important contributors to the antidepressant efficacy of serotonin modulating compounds. Serotonin receptors, including 5-HT$_{2A}$, are widely distributed throughout the gut and peripheral tissues (*Mawe & Hoffman, 2013*). Moreover, the serotonin produced in the gut accounts for more than 60% of peripheral serotonin in mice and more than 95% in humans (*Yano et al., 2015*). Enteric serotonin is predominantly secreted by enterochromaffin cells that line the gut. Thus, psychedelic drugs such as psilocybin have strong potential to influence enteric processes.

In addition to enteric factors, there is also substantial evidence that the gut microbiome has bi-directional effects on a variety of psychological disorders through a combination of neural, endocrine, and metabolic signals of the gut-brain axis (*Burokas et al., 2015*; *Carabotti et al., 2015*). The gut microbiome refers to the diverse array of microscopic organisms that exist in the gastrointestinal tract and their genomes. These microorganisms collectively contain a number of genes 150 times greater than that of the human genome (*Weinstock, 2012*). Gut bacteria such as *Bacilli*, *Bifidobacterium*, *Candida*, *Enterococcus*, *Escherichia coli* (*E. coli*), *Lactobacillus*, *Streptococcus*, and *Serratia* secrete serotonin, acetylcholine, dopamine, gamma-aminobutyric acid, glycine, and catecholamine (*Yano*

*et al., 2015*), which can promote serotonin production and release within the lining of the colon, affecting gut motility and permeability. Thus, gut microbes secrete a wide array of neurotransmitters and neuroactive molecules that regulate various complex cognitive processes, including mood, memory, learning, and cognition (*Yano et al., 2015*). Both preclinical and clinical studies have observed that gut microbiota affect the symptoms of mood disorders (*Cruz-Pereira et al., 2020*). Most prominent are studies using germ-free rodents, which have found that, when compared with specific pathogen free (SPF) counterparts free of certain infectious pathogens but not completely free of all microbes, germ-free rats develop anxiety-like behavior and germ-free mice develop exaggerated stress responses (*Crumeyrolle-Arias et al., 2014*; *Sudo et al., 2004*). These changes have been shown to be reversible upon recolonization of the gut through dietary probiotics (*Wallace & Milev, 2017*). These studies provide a basis for the idea that altering the gut microbiome could be an alternative therapeutic strategy to treat mood disorders. The exact relationships between bacterial populations and host remain relatively unknown due to the complexity of the gut-brain axis (*Dinan & Cryan, 2017*; *Sharon et al., 2016*).

Recent work on the importance of the gut microbiome suggests a need for more understanding of how therapeutics alter these microbe populations, potentially modulating a vast array of signaling and metabolic pathways. Prior studies of treatment with various traditional antidepressants have found inconsistent changes of gut microbiota diversity, richness, and composition (*Donoso et al., 2023*). Additionally, a study on ketamine, a novel fast-acting antidepressant, has reported dose-dependent relationships between drug treatment and shifts in gut microbe compositions in relatively short time periods (*Getachew et al., 2018*). Specifically, 7 days after a single ketamine treatment, some bacteria have an over 90-fold increase in abundance at the family level (*Getachew et al., 2018*). Combined, these studies suggest that psilocybin may also, in part, exert its antidepressant effects through similar mechanisms. This is particularly likely, since 5-$HT_{2A}$ receptors are an essential component of the gut-brain axis (*Fiorica-Howells et al., 2002*). Although gut microbiome has been proposed as a potential mechanism that psychedelics act upon (*Kelly et al., 2023*; *Kuypers, 2019*), no prior studies have investigated the effects of any psychedelics on gut microbe populations. Norbaeocystin is structurally similar to psilocybin and is also found in *Psilocybe* mushrooms. Prior studies have shown it does not cause head twitch behaviors in rats (*Adams et al., 2022*), a proxy for 5-$HT_{2A}$ activation and possibly hallucinations. Thus, by comparing psilocybin's effects to those of norbaeocystin, findings could contribute to the understanding of the role of 5-$HT_{2A}$ receptors in psilocybin's effects on the gut microbiome. Additionally, should norbaeocystin cause similar effects on the gut microbiome, this might suggest it also possesses therapeutic potential. Therefore, the primary objective of this study was to determine if psilocybin dose-dependently modulates gut bacterial composition. To accomplish this, animals were treated orally with varying dosages of psilocybin, vehicle, or norbaeocystin (a psilocybin precursor).

## MATERIALS AND METHODS

### Production of psilocybin and precursors from *E. coli*

Psilocybin and norbaeocystin-containing cell broths, acquired from Dr. J. Andrew Jones' lab at Miami University, were produced using a genetically modified *E. coli* biosynthetic production pathway (*Adams et al., 2022*; *Adams et al., 2019*). Concentrations of target metabolites in filtered cell broths were analyzed using HPLC. HPLC results indicated that the psilocybin containing broths had high levels of psilocybin (approx. 1 g/L), trace levels of norbaeocystin (<20 mg/L) and aeruginascin (<1 mg/L), and low levels of baeocystin (approx. 150 mg/L); while the norbaeocystin containing broths had high levels of norbaeocystin (approx. 1.5 g/L) only, with no baeocystin, psilocybin, or aeruginascin due to the lack of the methyltransferase responsible for the synthesis of the latter metabolites. The cell culture media broth contained none of the aforementioned metabolites.

### Animals, housing condition, treatment, and fecal collection

Thirty-nine adult male Long Evans rats (90–120 days of age) were bred in-house. Rats were housed individually in standard cages with water and a standard rodent chow (Purina Rodent Chow No. 5001, St. Louis, MO, USA) provided *ad libitum* and were kept on a 12 h:12 h light/dark cycle (lights on 0700) throughout the study. All the rats were SPF animals, as the cage bedding and materials were tested periodically for certain designated pathogens to ensure safety of researchers and research quality.

Rats were randomly assigned to one of 5 groups, which received oral gavage of low (0.2 mg/kg) or high (2 mg/kg) dosages of psilocybin (P-low and P-high, respectively), low (0.25 mg/kg) or high (2.52 mg/kg) dosages of norbaeocystin (N-low and N-high, respectively), or an equivalent volumetric amount of cell culture media broth as a negative control, with the order of administration randomized. As published previously, "low" dosages show no observable head twitch responses and "high" dosages cause observable head twitches in male Long Evans rats (*Adams et al., 2022*). This study was to determine if psilocybin or norbaeocystin (a psilocybin precursor) at different doses modulates gut bacterial composition. To accomplish this, gut bacterial composition caused by varying dosages of psilocybin or norbaeocystin with or without induced behavior were compared.

Fresh fecal samples were collected in the morning, in the order of defecation, but was random among groups. Collections occurred from 19 rats 1 week after treatment (control: $n = 7$; low dosage psilocybin: $n = 3$; high dosage psilocybin: $n = 2$; low dosage norbaeocystin: $n = 5$; and high dosage norbaeocystin: $n = 2$) and from 20 rats 3 weeks after treatment (control: $n = 8$; low dosage psilocybin: $n = 3$; high dosage psilocybin: $n = 4$; low dosage norbaeocystin: $n = 3$; and high dosage norbaeocystin: $n = 2$). All the collected fecal samples were snap frozen immediately and stored at −80 °C until processing. Each fecal sample was provided with a unique sample identification number. Thus, although researchers were aware of group allocation during drug administration and fecal sample collection, they were unaware of group allocation when samples were processed for gut microbiome analysis using 16S rDNA sequencing (see below for details).

All experimental interventions, including oral gavage and fresh fecal sample collection, were carried out in conscious rats without any anesthesia by experienced researchers. No rats showed any sign of distress throughout the study; thus, no analgesia was given. Consequently, all rats were included in this study and their results were reported. Criteria were established for euthanizing animals prior to the planned end of the experiment, but this was not needed. At the conclusion of the experiment, rats were euthanized with one IP injection of Euthasol (200 mg/kg body weight; a sodium pentobarbital-based drug). These euthanasia methods comply with AVMA standards. The research question, groups, fecal sample collection, and gut microbiome analysis using 16S rDNA sequencing were discussed before the study among involved researchers. All procedures were approved by Miami University's Institutional Animal Care and Use Committee (IACUC Project Number: 1033_2023_Apr).

## 16S rDNA sequencing and analysis

Genomic DNA was isolated and purified from the fecal samples *via* a commercialized kit (MPBio FastDNA™ Spin Kit for Feces SKU116570200, Santa Ana, CA, USA), and underwent PCR amplification of the 16S ribosomal DNA (rDNA) V4 region using the 515f/806r primer set (Earth Microbiome Project; http://www.earthmicrobiome.org/). Gel electrophoresis was then used to check the quality and size of the amplicons. Amplified 16S rDNA samples were purified by the SequalPrep Normalization Plate kit (Thermo Fisher, Waltham, MA, USA). Purified products were quantified by KAPA Library Quantification Kit Illumina Platforms (Kapa Biosystems, Wilmington, MA, USA), and used the Illumina Next Generation Sequencing MiSeq platform for amplicon sequencing at Miami University's Center for Bioinformatics and Functional Genomics for 16S rDNA sequencing based on our established protocol (*Xu et al., 2020*). The project is registered with the BioProject database (BioProjectID: PRJNA1054120). Raw sequencing data are accessible *via* http://www.ncbi.nlm.nih.gov/bioproject/1054120. Raw sequencing data were processed and cleaned. The sequencing reads of all samples were demultiplexed based on the associated GOLAY indices on the reverse primer. The generated sequences were analyzed using QIIME2 (*Xu, Yang & Zhu, 2020*). QIIME 2 is capable of analyzing MiSeq data with two or more biological replicates and tends to be conservative in revealing statistical significance (*Bolyen et al., 2019*), and has been used to analyze studies with sample sizes of 1 and 2 (*McKenzie et al., 2017*).

The sequences were grouped into 97% identity clusters *via* an operational taxonomic unit (OTU) selection method against the Greengenes reference database. Subsequently, the resultant feature tables were transformed into tables displaying relative abundance. The number and relative abundance of bacterial phylum level were exported and analyzed by GraphPad™ Prism 10. The taxonomic composition at the phylum level provides a broad framework for understanding the taxonomic diversity and organization of biological communities in ecological terms. Shannon diversity index considers the number of species indicating richness and their relative abundance indicating evenness, thus estimates the diversity of species in a community. The relative abundance refers to the percentage of one microbial phylum in relation to the total number of phyla in the community, which was
analyzed to indicate the population size of specific phyla and their commonalities with other phyla in the fecal samples.

## Statistical analysis

Significant differences in diversity at the phylum level and relative abundance of different bacterial populations between groups were determined by comparing the means of these variables between groups using a two-way analysis of variance (ANOVA). The False Discovery Rate (FDR) of all *post hoc* comparisons was controlled using the two-stage linear step-up method of *Benjamini, Krieger & Yekutieli (2006)* (GraphPad™ Prism 10, San Diego, CA, USA). Alpha levels for all comparisons were set at $p < 0.05$.

# RESULTS

## Effects of psilocybin and norbaeocystin on microbial ecology and diversity

Analysis of gut microbial taxonomic composition of bacterial communities at the phylum level across all treatment groups at both Week 1 and Week 3 timepoints revealed *Firmicutes* and *Bacteroidetes* as the dominant phyla, with *Proteobacteria*, *Actinobacteria*, *Tenericutes*, and *Verrucomicrobia* as sub-dominant phyla. Additionally, *Cyanobacteria* and *Deferribacteres* were present in some, but not all, samples. Furthermore, a few phyla, including *TM7*, *Elusimicrobia*, *Fusobacteria*, and *Lentisphaerae* contributing to the rare biosphere occurred at low abundance level within the community (Figs. 1A and 1B).

Microbial diversity measured by Shannon diversity index and analyzed by a two-way ANOVA (treatment × time) did not reveal any significant effect of psilocybin treatments ($F_{(2, 21)} = 0.7496$; $p = 0.4848$), time ($F_{(1, 21)} = 0.01407$; $p = 0.9067$), or their interaction ($F_{(2, 21)} = 0.3673$; $p = 0.6970$) (Fig. 1C). Similarly, a separate ANOVA found that microbial diversity was not significantly affected by norbaeocystin treatment ($F_{(2, 21)} = 0.3363$; $p = 0.7182$), time ($F_{(1, 21)} = 0.05253$; $p = 0.8209$), or their interaction ($F_{(2, 21)} = 0.5505$; $p = 0.5848$) (Fig. 1D). Therefore, neither psilocybin nor norbaeocystin at either dose resulted in any significant change in microbial ecology or diversity compared to the vehicle control (Fig. 1).

## Effects of psilocybin and norbaeocystin on microbial abundance of dominant phyla

The populations of two major gut microbiota phyla, *Firmicutes* and *Bacteroidetes* representing ~80% of gut microbiota (*Arumugam et al., 2011*; *Ley et al., 2008*), were analyzed.

*Firmicutes* abundance was analyzed by a two-way ANOVA (treatment × time), which revealed a significant interaction between psilocybin treatment and time ($F_{(2, 21)} = 3.797$; $p = 0.0391$), but *Firmicutes* abundance was not affected by main effects of psilocybin treatment ($F_{(2, 21)} = 0.1963$; $p = 0.8233$) or time ($F_{(1, 21)} = 0.0323$; $p = 0.8591$) (Fig. 2A). *Post hoc* comparisons indicated trends toward decreased *Firmicutes* abundance after high dose psilocybin treatment compared to the control group ($p = 0.0847$) and compared to low dose psilocybin treatment ($p = 0.0512$) at the Week 1 timepoint. This trend did not persist

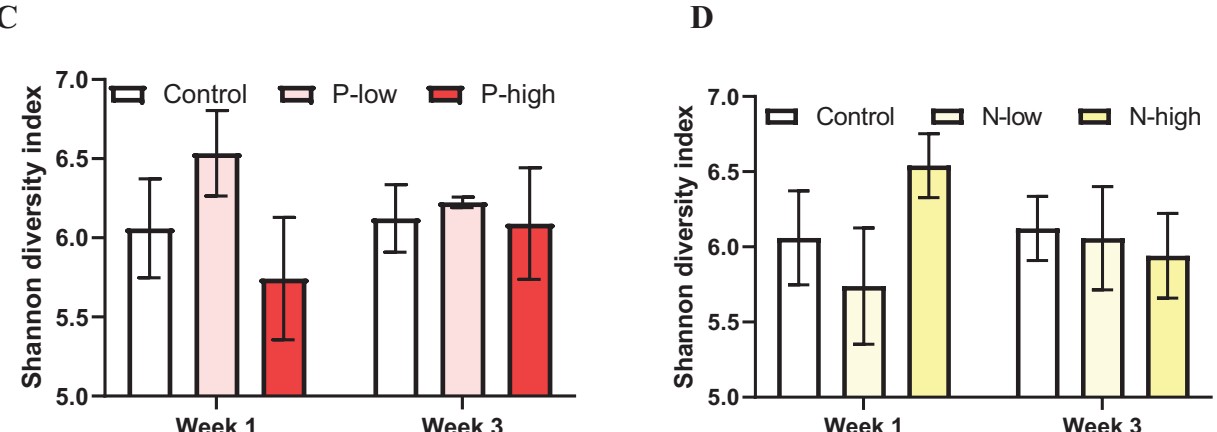

**Figure 1 Effects of psilocybin and norbaeocystin treatment on microbial diversity.** Effects of low dosage 0.2 mg/kg of psilocybin (P-low), high dosage 2 mg/kg of psilocybin (P-high), low dosage 0.25 mg/kg of norbaeocystin (N-low), and high dosage 2.52 mg/kg of norbaeocystin (N-high) on taxonomic composition of bacterial communities at the phylum level 1 week after treatment (A) or 3 weeks after treatment (B). Effects of P-low and P-high (C) and N-low and N-high (D) on microbial diversity, measured by Shannon diversity index. Significant differences in diversity at the phylum level between groups were determined by a two-way ANOVA followed by FDR-corrected *post hoc* comparisons analysis. $p < 0.05$ was considered statistically significant.

at the Week 3 timepoint, instead the trend reversed, showing increased *Firmicutes* abundance after high dose psilocybin treatment compared to the control ($p = 0.1282$) and low dose psilocybin treatment ($p = 0.1451$) (Fig. 2A). *Firmicutes* population was not significantly affected by norbaeocystin treatment ($F_{(2, 21)} = 0.7270$; $p = 0.4951$), time ($F_{(1, 21)} = 0.1179$; $p = 0.7347$), or their interaction ($F_{(2, 21)} = 0.8594$; $p = 0.4378$) (Fig. 2B). Therefore, psilocybin or norbaeocystin at either dose did not result in any significant changes in *Firmicutes* population compared to the vehicle control (Figs. 2A, 2B).

*Bacteroidetes* population was not significantly affected by psilocybin treatment ($F_{(2, 21)} = 2.430$; $p = 0.1124$), time ($F_{(1, 21)} = 0.1239$; $p = 0.7283$), or the interaction of treatment and time ($F_{(2, 21)} = 0.8893$; $p = 0.4259$) (Fig. 2C). Similarly, *Bacteroidetes* population was not significantly affected by norbaeocystin treatment ($F_{(2, 21)} = 1.732$; $p = 0.2013$), time ($F_{(1, 21)} = 0.06842$; $p = 0.7962$), or their interaction ($F_{(2, 21)} = 1.992$; $p = 0.1613$) (Fig. 2D).

The *Firmicutes/Bacteroidetes* ratio holds substantial clinical relevance, as its change has been implicated in various diseases, such as obesity, metabolic syndrome, systemic inflammation, cardiovascular disease, and inflammatory bowel disease (*Ley et al., 2006*; *Spychala et al., 2018*; *Stojanov, Berlec & Štrukelj, 2020*); thus, it is considered a critical factor influencing health. The *Firmicutes/Bacteroidetes* ratio was not significantly affected by psilocybin treatment ($F_{(2, 21)} = 1.562$; $p = 0.2331$), time ($F_{(1, 21)} = 0.6156$; $p = 0.4414$), or the interaction of treatment and time ($F_{(2, 21)} = 1.056$; $p = 0.3656$) (Fig. 2E). Similarly, the overall effect of norbaeocystin treatment on the *Firmicutes/Bacteroidetes* ratio was not significant ($F_{(2, 21)} = 0.8673$; $p = 0.4346$), nor was the effect of time ($F_{(1, 21)} = 0.4964$; $p = 0.4888$). However, a significant interaction between treatment and time was found ($F_{(2, 21)} = 3.651$; $p = 0.0436$). *Post hoc* comparisons indicated significantly increased *Firmicutes/Bacteroidetes* ratio after high dose norbaeocystin treatment compared to control ($p = 0.0140$) and after low dose norbaeocystin treatment ($p = 0.0140$) at Week 1 timepoint; such change did not persist at the Week 3 timepoint (Fig. 2F).

## Effects of psilocybin and norbaeocystin on microbial abundance of sub-dominant phyla

The populations of four minor gut microbiota phyla, *Proteobacteria*, *Actinobacteria*, *Tenericutes*, and *Verrucomicrobia* were analyzed.

*Proteobacteria* population at the phylum level was not affected by psilocybin treatment ($F_{(2, 21)} = 0.2846$; $p = 0.7552$), time ($F_{(1, 21)} = 0.6990$; $p = 0.4125$), or their interaction ($F_{(2, 21)} = 2.165$; $p = 0.1397$) (Fig. 3A). Analysis of *Proteobacteria* population revealed a significant interaction between norbaeocystin treatment and time ($F_{(2, 21)} = 3.946$; $p = 0.0351$), but not main effects of norbaeocystin treatment ($F_{(2, 21)} = 0.7901$; $p = 0.4668$) or time ($F_{(1, 21)} = 0.7484$; $p = 0.3968$) (Fig. 3B). *Post hoc* multiple comparisons indicated significantly decreased *Proteobacteria* abundance after low dose norbaeocystin treatment compared to control ($p = 0.0283$) and high dose norbaeocystin treatment ($p = 0.0282$) at Week 1 timepoint, but not at Week 3 timepoint (Fig. 3B).

*Actinobacteria* population at the phylum level was not affected by psilocybin treatment ($F_{(2, 21)} = 1.399$; $p = 0.2701$), time ($F_{(1, 21)} = 0.3574$; $p = 0.5567$), or their interaction

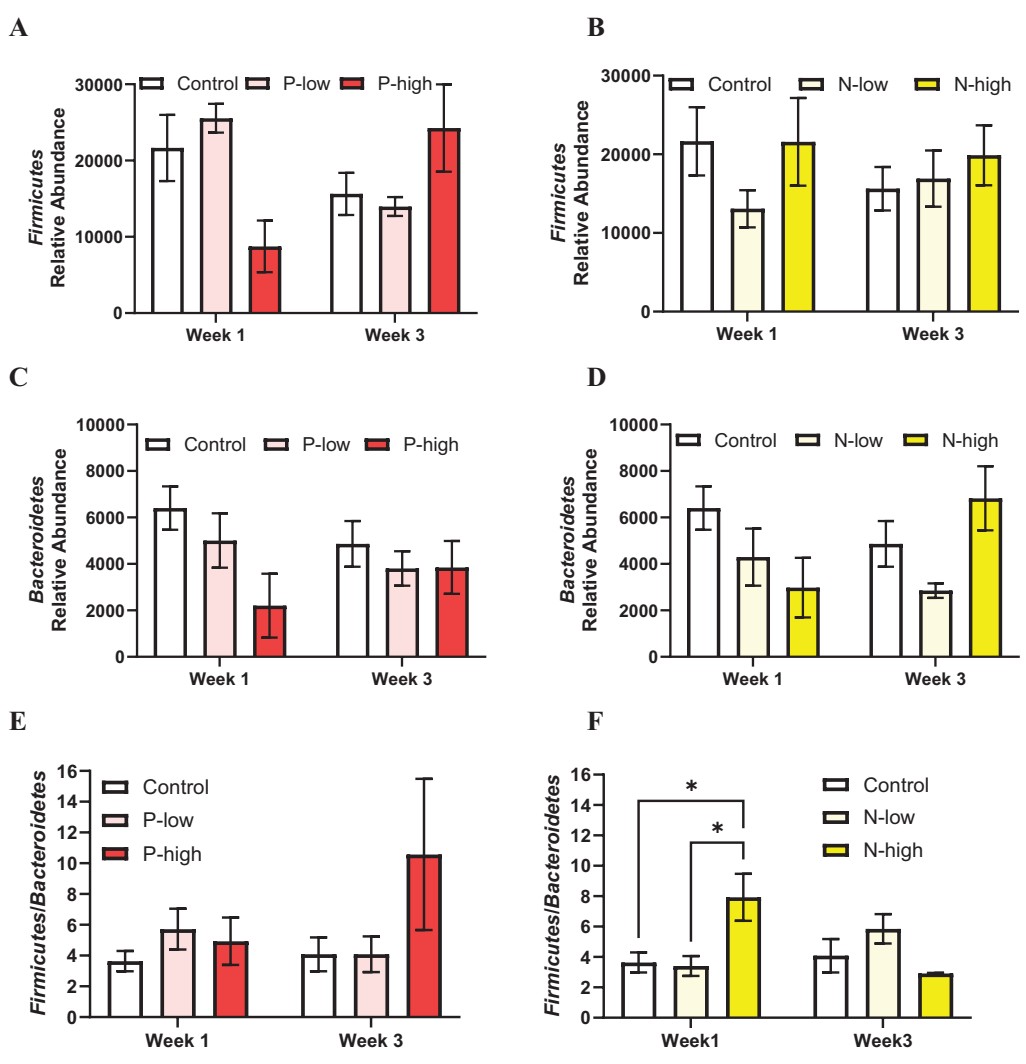

**Figure 2 Effects of psilocybin and norbaeocystin treatment on abundance of major microbial phyla at the phylum level and their ratio.** Effects of low dosage 0.2 mg/kg of psilocybin (P-low) and high dosage 2 mg/kg of psilocybin (P-high) (A), low dosage 0.25 mg/kg of norbaeocystin (N-low) and high dosage 2.52 mg/kg of norbaeocystin (N-high) (B) on *Firmicutes* abundance. Effects of P-low and P-high (C) and N-low and N-high (D) on *Bacteroidetes* abundance. Effects of P-low and P-high (E) and N-low and N-high (F) on the *Firmicutes/Bacteroidetes* ratio. Significant differences in relative abundance of *Firmicutes* or *Bacteroidetes* bacterial population and the *Firmicutes/Bacteroidetes* ratio between groups were determined by a two-way ANOVA followed by FDR-corrected *post hoc* comparisons analysis. $p < 0.05$ was considered statistically significant. *Indicated statistical significance.

($F_{(2, 21)} = 1.653$; $p = 0.2167$) (Fig. 3C). However, a significant main effect of norbaeocystin treatment ($F_{(2, 21)} = 3.631$; $p = 0.0452$) and interaction between norbaeocystin treatment and time ($F_{(2, 21)} = 4.977$; $p = 0.0176$) were revealed with no effect of time ($F_{(1, 21)} = 1.846$; $p = 0.1893$) (Fig. 3D). *Post hoc* comparisons indicated significantly increased *Actinobacteria* abundance after high dose norbaeocystin treatment compared to control ($p = 0.0014$) and low dose norbaeocystin treatment ($p = 0.0014$) at Week 1 timepoint. This change did not persist at the Week 3 timepoint (Fig. 3D). *Verrucomicrobia* population was

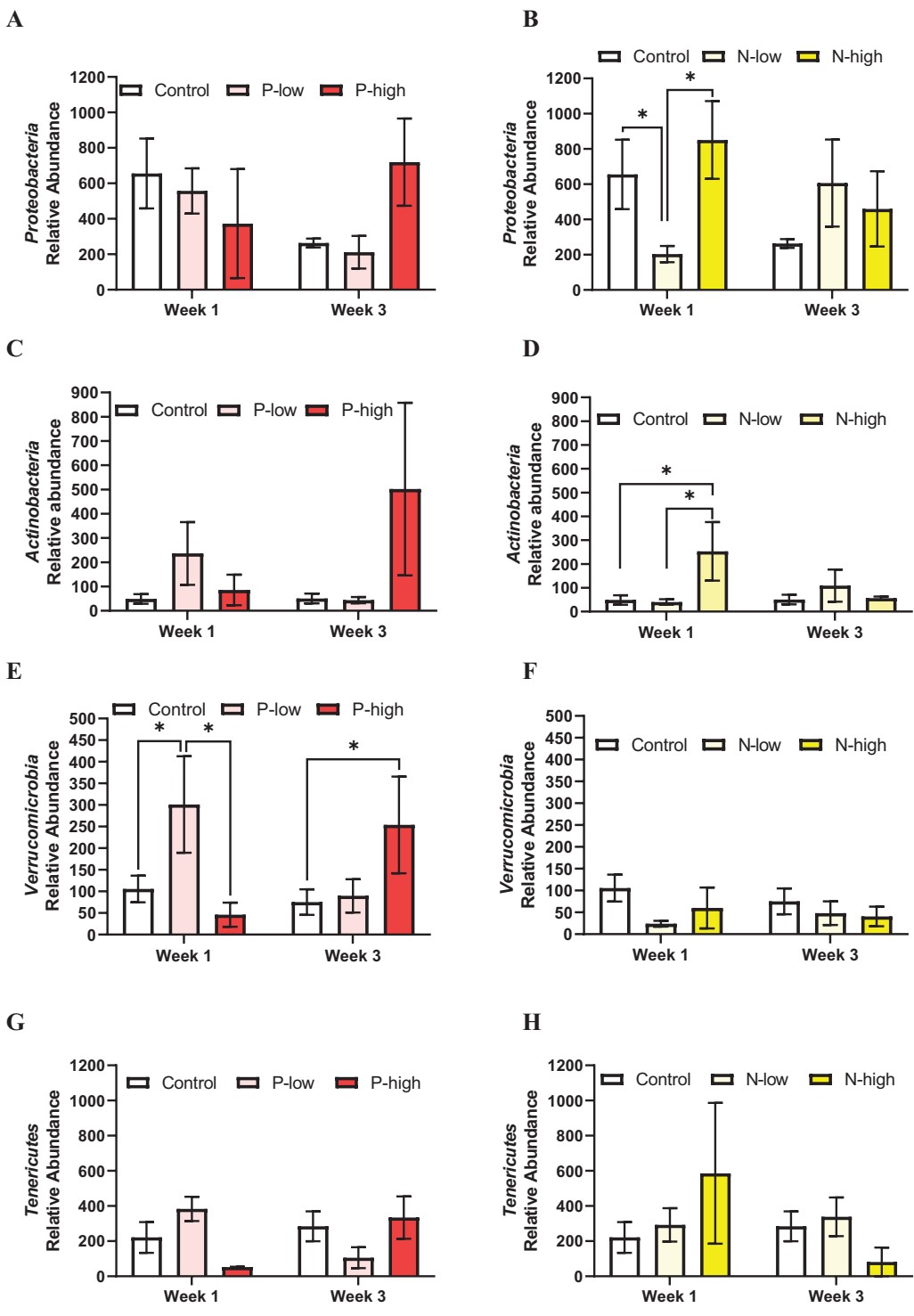

**Figure 3 Effects of psilocybin and norbaeocystin treatment on abundance of minor microbial phyla at the phylum level.** Effects of low dosage 0.2 mg/kg of psilocybin (P-low) and high dosage 2 mg/kg of psilocybin (P-high) (A), low dosage 0.25 mg/kg of norbaeocystin (N-low) and high dosage 2.52 mg/kg of norbaeocystin (N-high) (B) on *Proteobacteria* abundance. Effects of P-low and P-high (C) and N-low and N-high (B) on *Actinobacteria* abundance. Effects of P-low and P-high (E) and N-low and N-high (F) on *Verrucomicrobia* abundance. Effects of P-low and P-high (G) and N-low and N-high (H) on *Tenericutes*

**Figure 3** (continued)
abundance. Significant differences in relative abundance of bacterial population were determined by a
two-way ANOVA followed by FDR-corrected *post hoc* comparisons analysis. $p < 0.05$ was considered
statistically significant. *Indicated statistical significance.

significantly affected by the interaction of psilocybin treatment and time ($F_{(2, 21)}$ = 4.027;
$p$ = 0.0331); but there were not main effects of treatment ($F_{(2, 21)}$ = 1.646; $p$ = 0.2167) or
time ($F_{(1, 21)}$ = 0.04451; $p$ = 0.8349) (Fig. 3E). *Post hoc* comparisons indicated a significant
increase in *Verrucomicrobia* abundance after low dose psilocybin treatment compared to
control ($p$ = 0.0332) and high dose psilocybin treatment ($p$ = 0.0355) at the Week 1
timepoint; and a significant increase in *Verrucomicrobia* abundance after high dose of
psilocybin treatment compared to control ($p$ = 0.0288) at Week 3 timepoint (Fig. 3E).
*Verrucomicrobia* population at the phylum level was not changed by norbaeocystin
treatment ($F_{(2, 21)}$ = 1.735; $p$ = 0.2007), time ($F_{(1, 21)}$ = 0.07775; $p$ = 0.7831), or their
interaction ($F_{(2, 21)}$ = 0.3985; $p$ = 0.6763) (Fig. 3F).

*Tenericutes* population at the phylum level was not significantly affected by psilocybin
treatment ($F_{(2, 21)}$ = 0.1573; $p$ = 0.8554), time ($F_{(1, 21)}$ = 0.05945; $p$ = 0.8097), or their
interaction ($F_{(2, 21)}$ = 2.551; $p$ = 0.1019) (Fig. 3G); nor was it affected by norbaeocystin
treatment ($F_{(2, 21)}$ = 0.2521; $p$ = 0.7795), time ($F_{(1, 21)}$ = 1.386; $p$ = 0.2523), or their
interaction ($F_{(2, 21)}$ = 2.142; $p$ = 0.1424) (Fig. 3H).

## DISCUSSION

The gut-brain axis is a highly complex system, the importance of which is not yet fully
understood. Although neither psilocybin nor norbaeocystin treatments significantly
impacted gut microbe diversity (Fig. 1), some significant changes in microbial abundance
at the phylum level were observed. The Shannon diversity index was selected to assess
species diversity due to its robust, interpretable, and widely applicable nature, reflecting
both species richness and evenness within a community, which is well-suited for the
objectives of this study. A study comparing various commonly used diversity indices for
analyzing 16S gene sequencing data has reported that the Shannon diversity index is the
most effective measure (*Feranchuk et al., 2018*). Furthermore, the Shannon diversity index
is a widely used measure for quantifying species diversity within communities, making it
particularly valuable for comparing diversity levels among different studies. It is
noteworthy that alternative indices for assessing species diversity, such as Fisher's alpha
and abundance-based coverage estimator, could be explored in future studies to
comprehensively understand community diversity.

The human gut is predominantly composed of *Bacteroidetes* and *Firmicutes*,
complemented by sub-dominant *Actinobacteria*, *Proteobacteria*, and *Verrucomicrobia*
(*Qin et al., 2010*). Interestingly, neither *Firmicutes* nor *Bacteroidetes*, which together
represent ~80% of gut microbes (*Ley et al., 2008*), was significantly impacted (Fig. 2);
whereas three of the four sub-dominant bacterial phyla analyzed, *Proteobacteria*,
*Verrucomicrobia*, and *Actinobacteria* (*Ley et al., 2008*), were significantly impacted by
psilocybin or norbaeocystin treatments at different timepoints. Specifically, low dose

psilocybin treatment increased *Verrucomicrobia* abundance at the Week 1 timepoint and high dose psilocybin treatment increased *Verrucomicrobia* abundance at the Week 3 timepoint. Additionally, while low dose norbaeocystin decreased *Proteobacteria* abundance, high dose norbaeocystin increased *Actinobacteria* abundance, both of which occurred at the Week 1 timepoint (Figs. 3B, 3D). It is noteworthy that the rare biosphere encompasses a diverse array of microbial taxa. The specific composition of rare phyla may vary depending on factors such as environmental conditions, treatments, and the microbial community under study (*Hernandez et al., 2021*). Additionally, advancements in sequencing technologies and analytical methods may lead to the discovery of new rare taxa and provide further insights into the rare biosphere (*Jousset et al., 2017*; *Lynch & Neufeld, 2015*). In future investigations, with a large sample size and sufficient power for statistical analysis, a microbial network analysis could be pursued to elucidate microbial co-occurrence patterns within the community (*Banerjee, Schlaeppi & van der Heijden, 2018*). Such an analysis may yield valuable insights into the intricate relationships within microbiome data, enhancing our understanding of microbial interactions and their impact on community dynamics and function.

Emerging evidence supports the microbiota-gut-brain axis in regulation of physiology and behavior, and suggests that disturbance of the gastrointestinal microbiota could affect the immune system and psychiatric functioning (*Cruz-Pereira et al., 2020*). The bidirectional communication between gastrointestinal microbiota and immune system mediates many neural processes, such as neurogenesis, neurotransmission, neuroinflammation, and neurochemical functions such as activation of stress responses, depression, and other mental health disorders (*Cryan et al., 2019*; *Dinan & Cryan, 2015*; *Sarkar et al., 2018*). An interesting finding is that high dose norbaeocystin treatment increased the *Firmicutes*/*Bacteroidetes* ratio at the Week 1 timepoint (Fig. 2F). Findings from previous human and animal studies (*Bäckhed et al., 2004*; *Turnbaugh et al., 2009*) suggest the increased *Firmicutes*/*Bacteroidetes* ratio as a hallmark for obesity. However, all the rats in the current study were fed a standard rodent chow and were lean. Thus, findings from this study are not directly comparable to obesity studies in the literature. It has been reported that *Firmicutes* more effectively extract energy from food than *Bacteroidetes* (*Krajmalnik-Brown et al., 2012*). Thus, high dose norbaeocystin treatment may promote efficient absorption of calories, which awaits further investigation. In the current study, phyla *Verrucomicrobia* and *Actinobacteria* were increased by psilocybin and norbaeocystin, respectively. Phylum *Verrucomicrobia* are mucin-degrading bacteria, constitutes 3–5% of the bacterial community mainly residing in the intestinal mucosa that forms an interface between host and gut microbiome. Low abundance of *Verrucomicrobia* has been reported in prediabetic and type 2 diabetic patients (*Zhang et al., 2013*), in patients with inflammatory gut diseases such as Crohn's disease, ulcerative colitis, and inflammatory bowel disease (*Papa et al., 2012*; *Png et al., 2010*), and in populations with poorer sleep quality or disrupted sleep (*Anderson et al., 2017*). In contrast, abundance of *Verrucomicrobia* increases following dieting and Roux-en-Y gastric bypass in diabetic patients accompanied with many beneficial metabolic outcomes (*Barlow, Yu & Mathur, 2015*). Thus, a low level of *Verrucomicrobia* has been associated with metabolic disorders

and weakened immune system, while a high abundance of *Verrucomicrobia* is considered as a potential biomarker of a healthy gut status (*Anderson et al., 2017*; *Barlow, Yu & Mathur, 2015*; *Papa et al., 2012*; *Png et al., 2010*; *Zhang et al., 2013*). Phylum *Actinobacteria* contributes to the maintenance of gut homeostasis and supports immune system (*Binda et al., 2018*). In contrast to beneficial phyla *Verrucomicrobia* and *Actinobacteria* that were increased following treatment of psilocybin and norbaeocystin, respectively, high abundance of phylum *Proteobacteria* is considered as a microbial signature of disease (*Rizzatti et al., 2017*) and was decreased by low dose norbaeocystin treatment. Thus, it is possible that psilocybin and norbaeocystin could be candidates for alleviating gut dysbiosis and producing positive effects in disease conditions.

The findings of the current study are promising, as significant alteration of the gut microbiome may provide a possible explanation as to why psilocybin users report a reduction of depressive symptoms after treatment (*Chen et al., 2023*; *Meccia, Lopez & Bagot, 2023*). It has been proposed that psychedelics may affect gut microbiome to influence their treatment responses (*Kelly et al., 2023*; *Kuypers, 2019*). To our knowledge, the current study is the first study that investigated the effects of the tryptamines, psilocybin and norbaeocystin, on gut microbe populations. Unlike conventional mood modulating drugs that require chronic doses over a long timeline, it is possible that psilocybin in part works by altering microbe populations within the gut, potentially targeting a component of the disease state rather than treating the symptoms. These results also suggest that norbaeocystin, a psilocybin precursor with limited study in the peer reviewed literature (*Adams et al., 2022*), may warrant further investigation as a potential antidepressant. Additionally, in future investigations, the exploration of microbial and functional biomarkers in the microbiome affected by psilocybin or norbaeocystin treatments could be pursued. For example, conducting linear discriminant effect size analysis, a methodology that facilitates the inference of functional and metabolic potential from microbial community metagenome (*Segata et al., 2011*) may offer valuable insights.

The limitation of this study is low sample sizes analyzed for some groups. Although QIIME 2 is capable of analyzing MiSeq amplicon data with two or more biological replicates and tends to be conservative in revealing statistical significance (*Bolyen et al., 2019*), as shown in a publication with some sample sizes of 1 and 2 (*McKenzie et al., 2017*), we should interpret findings with caution. The lack of statistical significance in diversity and some phylum abundance, along with high variability of the relative abundance of some phyla, could be due to the small sample size of this preliminary study. Consequently, the findings from this preliminary study have limited generalizability, and would require further validation with larger sample sizes and comparing across routes of drug delivery.

One caveat to the current work is that it was conducted in normal, healthy rats. It is likely that rats modeling a disease-state, such as chronic stress, anxiety, and depression, may respond differently to psilocybin and/or norbaeocystin. Previously, we have reported the impact of chronic stress on gut microbiome diversity and composition, leading to gut dysbiosis (*Xu et al., 2020*). Rats with disturbed microbiome may react very differently to these drugs. As such, findings from healthy rats in this study may not generalize to other animals or treatment conditions. Further research is needed using disease models, where

multiple physiological, biochemical behavioral and microbiome outcomes are evaluated, such that biological mechanisms can be elucidated. Additionally, there is still little information on how psilocybin and norbaeocystin interact with the body, and continued study is needed in order to inform potential side effects in human trials. Altogether, psilocybin and norbaeocystin stand as strong candidates for managing gut dysbiosis.

## CONCLUSIONS

In order to use psilocybin to treat mood disorders, it is critical to better understand its efficacy and safety. However, the schedule I status of psilocybin has greatly hindered advancements in research. Psilocybin may also be used to treat other diseases, such as those related to gut health. For example, the FDA recently approved a Phase 2A clinical trial for the treatment of irritable bowel syndrome with psilocybin. Although our study does not use a paradigm that induces stress or depression, nor does it determine psilocybin's ability to modulate mood *via* the gut-brain axis, it does begin to probe the mechanisms by which psilocybin affects body physiology and behavior. The observed alterations to the gut microbiome show promise for the ability of psilocybin and norbaeocystin to affect the gut microbiome in a positive manner and establish a path for future research to investigate how psilocybin, or other related tryptamines, could be used to modulate the gut microbiota to treat dysbiosis as well as other disorders. Prior to this study the potential effects of psilocybin and its biosynthetic precursor, norbaeocystin, on gut microbe populations was unknown. Further investigations building upon this work could open the door to a new potential avenue for pharmaceuticals which target the gut-brain axis.

## ACKNOWLEDGEMENTS

The authors would like to thank Miami University graduate students involved in sample collection.

### Funding

This work was supported by a sponsored research grant from PsyBio Therapeutics (J. Andrew Jones, Matthew S. McMurray) and is supported by Miami University Faculty Research grant (Haifei Shi). There was no additional external funding received for this study. The funders had no role in study design, data collection and analysis, decision to publish, or preparation of the manuscript.

### Grant Disclosures

The following grant information was disclosed by the authors:
PsyBio Therapeutics.
Miami University Faculty Research grant.

## Competing Interests

J. Andrew Jones is a significant stakeholder at PsyBio Therapeutics. PsyBio Therapeutics has licensed tryptamine biosynthesis-related technology from Miami University. J. Andrew Jones and Matthew S. McMurray are co-inventors on several patent applications related to tryptamine biosynthesis and the impacts of tryptamines on animal behavior. All other authors declare no conflicts of interest.

## Author Contributions

- Mengyang Xu conceived and designed the experiments, performed the experiments, analyzed the data, prepared figures and/or tables, authored or reviewed drafts of the article, and approved the final draft.
- Andor J. Kiss conceived and designed the experiments, performed the experiments, analyzed the data, prepared figures and/or tables, authored or reviewed drafts of the article, and approved the final draft.
- J. Andrew Jones conceived and designed the experiments, performed the experiments, analyzed the data, prepared figures and/or tables, authored or reviewed drafts of the article, and approved the final draft.
- Matthew S. McMurray conceived and designed the experiments, performed the experiments, analyzed the data, prepared figures and/or tables, authored or reviewed drafts of the article, and approved the final draft.
- Haifei Shi conceived and designed the experiments, performed the experiments, analyzed the data, prepared figures and/or tables, authored or reviewed drafts of the article, and approved the final draft.

## Animal Ethics

The following information was supplied relating to ethical approvals (*i.e.*, approving body and any reference numbers):

All procedures were approved by Miami University's Institutional Animal Care and Use Committee (IACUC Project Number: 1033_2023_Apr).

## DNA Deposition

The following information was supplied regarding the deposition of DNA sequences:

The raw sequencing data are available at Bioproject: PRJNA1054120.

http://www.ncbi.nlm.nih.gov/bioproject/1054120

## Data Availability

The Shannon diversity and phylum of samples are available in the Supplemental Files.

## Supplemental Information

Supplemental information for this article can be found online at http://dx.doi.org/10.7717/peerj.17517#supplemental-information.

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
