# Peer review of "Effect of oral tryptamines on the gut microbiome of rats—a preliminary study"

_PeerJ, doi:10.7717/peerj.17517_

## Round 0.1 · original submission · Major Revisions

Please reply in particular to the queries of reviewer 4.

·

Basic reporting

In this manuscript, the authors have reported “Effect of oral tryptamines on the gut microbiome of rats – a preliminary study''. The work, as with all work coming from this particular research group, is fundamentally sound. The existing literature is well cited. The reviewer recommends this manuscript for publication in PeerJ journal.

Experimental design

The paper is well written and easy to follow. Significantly, this work could help scientific readers and could open the door to a new potential avenue for pharmaceuticals which target the gut-brain axis.

Validity of the findings

Overall, the paper is well documented, easy to read, and it contains a lot of comprehensive information. I suggest to the editor to be willing to consider it in the current version for publication.

Additional comments

I suggest to the editor to be willing to consider it in the current version for publication.

Reviewer 2 ·

Basic reporting

Dear Authors,

Your manuscript titled “Effect of oral tryptamines on the gut microbiome of rats – a preliminary study” by Mengyang Xu et al. provides intriguing insights into the role of psilocybin and norbaeocystin on gut microbiome dynamics and their potential antidepressant mechanisms.However, there are several grammatical mistakes and figure legends are not properly written.

Line 25: The phrase "varying dosages" could be more specific. It might be beneficial to briefly mention the range of dosages used for clarity, even in an abstract.

Line 28: The construction "either psilocybin or norbaeocystin did not significantly affect" might read more smoothly as "neither psilocybin nor norbaeocystin significantly affected."

Line 43: The phrase "a single dose of the tryptamine" could be clarified as "a single dose of psilocybin" for direct reference.

Line 44 List a few traditional antidepressants for non-familiar reader.

Line 62: "Specifically, substantial research has shown" could be streamlined to "Significant research indicates" for conciseness.

Line 65 avoid repetition of "and." In the sentence …….the gut and peripheral tissues (Mawe & Hoffman 2013) and the serotonin produced in the gut accounts for more than 60%.........

Consider changing to

the gut and peripheral tissues (Mawe & Hoffman 2013). Moreover, the serotonin produced in the gut accounts for more than 60%.........

Line 68: "Thus, psychedelic drugs have strong potential to influence….." The term "psychedelic drugs" is broad; specifying that this includes tryptamines discussed in the study could clarify the focus.

Line 73: The clarification "(i.e., microbiota)" might be unnecessary since "microbiome" is a well-understood term in this context.

Line 80: "as well as" could be simplified to "and" for conciseness.

Line 81: Consider splitting the long list of cognitive processes with semicolons for clarity, e.g., "mood; memory; learning, and cognition."

Line 105: The ending ", however" is unnecessary and can be removed for a smoother sentence flow.

What do you mean by P-low and P-high in figures. Explain in legends and text.

Figure 1: "Effects of psilocybin and norbaeocystin treatment on microbial diversity." Provide an elaborate figure legend.

Figure 3 There is no * indicating P value in main figure. Also indicate which statical test you used to calculate p value in legends.

In summary, while the manuscript offers valuable data on the effects of tryptamines on the gut microbiome, attention to detail in the presentation of dosage ranges, statistical significance, and definitions will substantially improve the manuscript’s impact and readability. I look forward to the revised version of your work.
Best regards,

Experimental design

'no comment'

Validity of the findings

'no comment'

Additional comments

Dear Authors,

Your manuscript provides an intriguing look at how tryptamines affect the gut microbiome and their potential antidepressant mechanisms. To strengthen your submission, consider detailed proofreading to polish language use and clarify scientific terms.

Reviewer 3 ·

Basic reporting

The study was well reported with clear and well-written English

Experimental design

The methodology used in the study is soundly described and can be replicated.

Validity of the findings

The findings were well reported and nicely presented in the figures. The supplementary data provided were also valuable.

Additional comments

The study by Xu et al. is a preliminary study on the Effect of oral tryptamines on the gut microbiome of rats. This manuscript is generally well-written and well-presented. The methodology is also sound and adequately described. However, I have a minor concern before the paper can be accepted for publication.
1. Some scientific names in the manuscripts are not correctly written. For example, check line 117, E. coli.
2. I wonder why the author did not provide information on the phylogenetic diversity and relatedness of the microbes, especially the dominant taxa, in the manuscript.

·

Basic reporting

The authors present an article on evaluating an antidepressant drug and its effect on the intestinal microbiota of healthy rats. They have a good level of English. However, not all their references are updated, and they do not demonstrate a good understanding of the microbial ecology of the intestinal microbiota of mammals.

Experimental design

The research question of this manuscript is interesting. However, this study does not have rigorous research and does not have validation from a bioethics committee to carry out the experiments.
Furthermore, they do not describe the bioinformatics workflow they followed with the qiime2 software.

Validity of the findings

This manuscript may have promising discoveries, but the authors are not performing relevant statistical tests and are not fully explaining everything they are obtaining with their data. In recent years, a comparison at the phylum level has not said much about the community.
Furthermore, they are not adequately discussing their work in terms of a microbial ecology study.

Additional comments

Detailed comments and suggestions are in the attached file. Please revise them

---

## Round 0.2 · accepted · Accept

The manuscript can be accepted as the authors replied satisfactorily.